# The STAT3-Regulated Autophagy Pathway in Glioblastoma

**DOI:** 10.3390/ph16050671

**Published:** 2023-04-29

**Authors:** Ronald Nicholas Laribee, Andrew B. Boucher, Saivikram Madireddy, Lawrence M. Pfeffer

**Affiliations:** 1Department of Pathology and Laboratory Medicine, The Center for Cancer Research, College of Medicine, University of Tennessee Health Science Center, Memphis, TN 38163, USA; rlaribee@uthsc.edu; 2Department of Neurosurgery, College of Medicine, University of Tennessee Health Science Center, Memphis, TN 38163, USA; aboucher@semmes-murphey.com; 3College of Medicine, University of Tennessee Health Science Center, Memphis, TN 38163, USA; smadired@uthsc.edu

**Keywords:** autophagy, STAT3, glioblastoma, GBM, GBM cancer stem cell, GSC, glioma

## Abstract

Glioblastoma (GBM) is the most common primary brain malignancy in adults with a dismal prognosis. Despite advances in genomic analysis and surgical technique and the development of targeted therapeutics, most treatment options are ineffective and mainly palliative. Autophagy is a form of cellular self-digestion with the goal of recycling intracellular components to maintain cell metabolism. Here, we describe some recent findings that suggest GBM tumors are more sensitive to the excessive overactivation of autophagy leading to autophagy-dependent cell death. GBM cancer stem cells (GSCs) are a subset of the GBM tumor population that play critical roles in tumor formation and progression, metastasis, and relapse, and they are inherently resistant to most therapeutic strategies. Evidence suggests that GSCs are able to adapt to a tumor microenvironment of hypoxia, acidosis, and lack of nutrients. These findings have suggested that autophagy may promote and maintain the stem-like state of GSCs as well as their resistance to cancer treatment. However, autophagy is a double-edged sword and may have anti-tumor properties under certain conditions. The role of the STAT3 transcription factor in autophagy is also described. These findings provide the basis for future research aimed at targeting the autophagy-dependent pathway to overcome the inherent therapeutic resistance of GBM in general and to specifically target the highly therapy-resistant GSC population through autophagy regulation.

## 1. Introduction

### 1.1. GBM

Gliomas are the most common primary intracranial neoplasms in adults, and they are a leading cause of cancer-related morbidity and mortality in the United States [1]. Based on morphologic evidence of differentiation along astrocytic, oligodendroglial, or mixed lineages, a grading system has been developed by the World Health Organization to classify gliomas; it was most recently updated in 2021 [2]. Gliomas are rated on a scale of 1 to 4 in terms of aggressiveness with glioblastoma (GBM) being the most aggressive subtype of grade 4 gliomas. Gliomas graded as 3 and 4 are collectively known as high-grade gliomas that have a relatively poor prognosis, whereas grades 1 and 2 are regarded as low-grade gliomas and are associated with a slower growth rate and better patient survival (3–8 years). Invasion and neoangiogenesis are important hallmarks of GBM, which account for nearly 75% of all gliomas. These attributes make GBM a highly lethal cancer that has a median survival of 10–15 months [1]. In the United States alone, roughly 10,000 cases of GBM are reported each year, and GBM accounts for nearly 75% of all diagnosed gliomas [3]. Although advances in surgical resection, adjuvant chemotherapy, and radiation therapy have slightly improved disease course and outcome for patients with GBM [4], the 5-year overall prognosis for GBM is dismal (<10% survival), and it has remained unchanged for decades [1]. Another contributing factor to poor patient outcomes is that the overall median time for GBM tumor recurrence after surgery is ~7 months, and patients with recurrent GBM have a bleak prognosis [5]. Despite advances in GBM molecular characterization and targeted therapies, the blood–brain barrier renders most treatment options in GBM ineffective by hindering the crossing of therapeutic agents into the brain.

As the historic name glioblastoma multiforme indicates, GBM is a very heterogenous tumor that is comprised of multiple cell types. A prominent cell type within GBM tumors is glial cells, which share molecular and genetic properties with other glial tumors, including oligodendroglioma and other astrocytomas [6,7]. However, GBM also contains other cell types, including oligodendrocyte progenitor cells, pericytes, non-tumor endothelial cells, macrophages, and other immune cells. These diverse cells are at distinct stages of proliferation and differentiation that are determined mainly by differences in the genetic expression, localization, and activation of various cell signaling pathways [8,9]. Furthermore, within GBM tumors is a distinct subpopulation of tumor-initiating cells that resemble neural progenitor cells, and hence they are designated GBM cancer stem-like cells (GSCs). GSCs express various neural stem cell markers, such as Nestin, CD133, CD44, and Sox2; they can migrate and invade into normal brain tissue; and they can self-renew and differentiate into multiple cell types [10]. Most importantly, GSCs have a high tumor-initiating capacity and intrinsic therapeutic resistance, which is believed to drive GBM tumorigenesis by repopulating the tumor after surgery, adjuvant radiotherapy, and chemotherapy [11]. GSCs are very difficult to target because they are not a static population of cells, and they may be at various states of differentiation and sensitivity to therapeutics. For example, the angiogenic factor apelin A (also called ELABELA) was found to be highly expressed in various GSCs, but its expression was found to vary markedly within GSCs isolated from different regions on the same brain tumor [12]. Moreover, classical neural stem marker expression in GSCs varies significantly in different PDXs derived in the same laboratory. For example, in our laboratory, X6 GSCs are predominantly CD133^high^/CD44^high^, while X10 and X16 GSCs are mainly CD133^low^/CD44^high^. Moreover, X6 GSCs are predominantly Nestin^high^/Sox2^high^, while X10 and X16 GSCs are Nestin^high^/Sox2^low^. Since GSCs vary dramatically in their expression of stem cell markers, these cells cannot be designated as GSCs based solely on the expression of such markers.

Gene expression analysis of GBM patient samples in The Cancer Genome Atlas (TCGA) database resulted in the classification of GBM into three general transcriptional subtypes: proneural, classical, and mesenchymal [13]. However, single-cell genomic analysis demonstrated that multiple molecular subtypes exist within an individual GBM tumor, and gene expression varies dramatically across individual tumor cells, thus providing further evidence that there is high molecular heterogeneity in GBM tumors [14]. Furthermore, despite the development of targeted therapeutics based on thorough genomic analysis, all GBM patients are still treated with nearly the same therapeutic regimen, which has remained unchanged for decades. A hallmark of GBM is the activation of receptor kinase signaling pathways such as the epidermal growth factor receptor (EFGR) or the platelet-derived growth factor receptor (PDGFR), due to genomic alterations including activating mutations, amplification, or mutation [15]. This genetic deregulation hyperactivates the PI3K-Akt-mTORC1 signaling axis that is a common genetic driver for GBM. For example, the EFGR pathway is genetically altered in ~50% of the GBM patient samples analyzed [16,17]. The EGFR variant III (EGFRVIII), formed by the deletion of exons 2–7, is the most common EGFR variant in high-grade gliomas and results in a constitutively active kinase whose activity is independent of ligand binding [18,19]. Increased EGFR signaling causes the aberrant activation of EGFR activated signaling pathways, including STAT3 and the PI3K/AKT/mTOR signal pathway, which control cell proliferation, angiogenesis, invasion, and apoptosis.

### 1.2. Clinical Therapy for GBM

Due to its relatively high recurrence rate, the physical location of the tumor in the brain parenchyma, the highly invasive nature of the tumor into adjacent normal brain, and the diversity of cell types within the tumor, GBM is extremely difficult to treat, and nearly all patients eventually succumb to the disease. Initial treatment includes maximal surgical resection of the mass, with subsequent focal radiation therapy and chemotherapy treatment with the alkylating agent temozolomide (TMZ) [19,20]. The complete resection of all tissue is impossible given the microscopic invasive nature of GBM and its proximity to eloquent structures that control speech, motor and sensory function, and other critical bodily functions. The tumor cells that remain after surgery contribute to the high resistance and tumor recurrence rates. It was only in recent decades that chemotherapy began to be included in the standard regimen for newly diagnosed GBM, with TMZ first being granted FDA approval in 2005 [20]. Clinical trials of TMZ and radiotherapy following surgical resection was shown to prolong median survival periods by up to 16 months [20,21,22]. Currently, carmustine, another alkylating agent, and bevacizumab, an anti-angiogenic agent targeting the vascular endothelial factor, are also FDA approved for GBM and are reserved for the treatment of recurrent GBM [23,24]. However, even after initial surgical resection, chemotherapy, and radiation, the survival of patients with GBM is only slightly prolonged and is mainly palliative. The current standard of care for GBM patients has not been significantly revised since 2005 [11,20,25]. 

## 2. The STAT3 Signaling Pathway

### 2.1. The STAT3 Signaling Pathway in GBM

Several oncogenic signaling pathways, including the EGFR pathway that is deregulated in GBM, contribute to GBM progression by converging on the important STAT3 molecular hub that controls critical biological functions, including cell proliferation, differentiation, survival, angiogenesis, and the tumor immune response [10,26,27,28,29,30,31,32,33,34]. STAT3 is activated through its phosphorylation by a wide variety of cytokines and growth factors [34,35,36]. High STAT3 activation is found in many GBM tumors [37,38], and STAT3 signaling actively participates in GBM tumor formation and progression [38]. Furthermore, STAT3 activation is highly elevated in GSCs that were isolated from various GBM patient tumors [39], and STAT3 activation is critical for GSC proliferation in vitro and the formation of xenografts in immunocompromised mouse models [10]. STAT3 promotes the expression of pro-tumorigenic genes in GSCs, which are involved in cell cycle progression, the remodeling of the extracellular matrix, and the expression of genes that encode cytokines and growth factors [10].

Many intracellular and receptor-associated tyrosine kinases phosphorylate STAT3 on tyrosine (Y)-705 (STAT3 Y705ph) to induce the formation of STAT3 homodimers or heterodimers with other STAT proteins. The STAT protein family consists of STAT1, STAT2, STAT3, STAT4, STAT5a, STAT5b, and STAT6. Upon phosphorylation, STAT3 dimerizes through the binding of the phosphorylated tyrosine residue in STAT3 to the SH2 domain of STAT proteins. These dimers then translocate into the nucleus and regulate gene transcription by binding to the promoters of STAT-regulated genes. For example, the cytokines interferon-alpha/beta and interferon-gamma signal through the type I and type II interferon receptor, respectively, to induce rapid and transient STAT3 activation through the JAK (Janus) tyrosine kinase pathway [33]. However, many upstream tyrosine kinases can phosphorylate STAT3 [40]. STAT3 also undergoes serine (S)-727 phosphorylation, which modulates gene transcription and translation, and mitochondrial function [41]. STAT3 activity is highly regulated at multiple levels to ensure proper biological function. 

Under normal physiological conditions, STAT3 Y705ph is transient, and its dephosphorylation is tightly controlled. However, aberrantly elevated STAT3 activity has been estimated to occur in more than 70% of human cancers [42]. Serine and tyrosine phosphorylation of STAT3 has been shown to be evident in established GBM cell lines, GSCs, and GSCs induced to undergo differentiation [10,43,44]. Furthermore, STAT3 phosphorylation plays a critical role both in maintaining GSCs in their stem-like state and their high intrinsic tumorigenicity [10]. RNA sequencing and microarray analysis identified STAT3-regulated genes in GSCs, including genes that were classical JAK-STAT signaling-pathway-induced genes. Many STAT3-regulated genes are overexpressed in GBM and are associated with poor patient survival [10]. Furthermore, high STAT3 expression is associated with increasing glioma grade and poor prognosis in GBM. 

### 2.2. Therapeutic Targeting of STAT3

Pharmacologic inhibitors of STAT3 phosphorylation reduce several hallmarks of cancer, including GBM cell proliferation, survival, migration, invasion, and tumorigenicity [43]. In addition, the silencing of STAT3 expression by shRNA in GSCs showed that STAT3 plays a key role in STAT3-regulated gene expression in GSC tumorigenesis [10]. For example, knockdown of STAT3 suppressed the expression of genes in GSCs that are involved in cell proliferation, while inducing the expression of pro-apoptotic genes. Most pertinent for this review was the finding that in GSCs, STAT3 regulated the expression of the pro-autophagy gene *ATG5* [10]. 

Because of their pro-tumorigenic role, STAT3 inhibitors have been developed as anticancer agents in various human cancers [45]. In fact, the STAT3 inhibitor WP1066 is presently in a clinical trial in GBM (NCT01904123). In a recent study, a rationally designed novel small-molecule inhibitor of STAT3 Y705ph was identified, which appears to be significantly more potent than WP106 [46]. This drug, denoted SS-4, modulates STAT3-regulated gene expression, induces apoptosis, inhibits cell proliferation and invasion, and reduces the growth GBM intracranial tumor xenografts [46]. An underappreciated function of STAT3 is its role in autophagy and lysosomal function, which will be discussed in depth later in this review. 

In addition to its direct effects on tumor cells, STAT3 may play an important role in the regulation of the tumor microenvironment (TME) in GBM. While blood cancers elicit a strong inflammatory response (designated “hot” cancers), many solid tumors, including GBM, evade the immune response and are considered “cold” tumors [47]. Although genomic analysis of GBM tumors demonstrates the infiltration of diverse immune cells [48], the immune response in GBM is blunted. STAT3 has been shown to play a role in regulating the anti-inflammatory response in immune cells by stimulating the expression of genes that suppress the transcription of proinflammatory genes, anti-inflammatory chemokines, and cytokines. However, there is a major gap in our understanding of the mechanism by which STAT3 regulates these proinflammatory and anti-inflammatory genes to suppress GBM tumor immune responses. For example, interleukin 6 (IL-6) is an important STAT3 activator and a major mediator of inflammation that acts on tumor cells to induce the expression of STAT3 target genes [49]. STAT3 also binds to the promoter of the IL-6 gene, thereby increasing IL-6 gene expression and producing a positive feedback IL-6/JAK/STAT3 loop [50]. However, the pharmacologic or genetic inhibition of STAT3 increases IL-6 expression in GBM cells and GSCs through unclear mechanisms, suggesting that therapeutically muting STAT3-dependent IL-6 induction may inadvertently suppress the tumor immune response to GBM. 

Immune checkpoint proteins, including cytotoxic T-lymphocyte-associated protein 4 (CTLA4), programmed death (PD-1), and programmed death ligand 1 (PDL-1), also play critical roles in immune suppression, especially anti-tumor immunity. The IFN-responsive gene PD-L1 is expressed on multiple types of immune cells and many various cancers including GBM. Although the blockade of PD-1/PD-L1 interaction has emerged as an effective therapeutic strategy for enhancing anti-tumor immune responses [51], many solid tumors such as GBM do not respond or progress following therapy targeting this interaction. Sensitization to immune checkpoint blockage is associated with the activation of STAT1 signaling and IFN-responsive gene signature in the TME [52]. Improving the treatment of brain tumors requires a more complete understanding of the properties and functions of the cells in the TME. For example, glioma-infiltrating microglia and macrophages reportedly comprise ~30% of GBM tumor cell mass; however, these cells produce cytokines and growth factors that promote tumor growth and an immunosuppressive microenvironment [53].

## 3. Autophagy

### 3.1. General Review of Autophagy

Maintaining metabolic homeostasis requires adapting the cell’s metabolic and proliferative requirements to the nutrient and energy resources available to support these processes. The dysregulation of these mechanisms causes cellular stress that, if left unattended, impairs cellular function and can cause disease [54]. Among the most critical homeostatic mechanisms that all eukaryotic cells employ is autophagy. This process, at its simplest level, involves the digestion of intracellular constituents with the ultimate goal of recycling the resulting nutrients to maintain cell metabolism. As such, autophagy is an essential coping mechanism for cells undergoing not only nutrient stress but also a wide variety of other extracellular and intracellular stressors that can impinge on cellular function [55]. Furthermore, cells utilize constitutively low levels of autophagy as a quality control mechanism to remove protein aggregates and dysfunctional organelles to prevent the impairment of cell fitness [56]. Given autophagy’s critical role in maintaining cellular homeostasis, defective autophagy is a known contributor to many diseases, including neurodegenerative disease [56]. However, in the context of cancer, autophagy functions early on as a tumor-suppressive mechanism by promoting the clearance of potentially oncogenic proteins or dysfunctional organelles. During the later stages of tumorigenesis, elevated autophagy can promote cancer cell survival within the metabolic restrictive tumor microenvironment, it can facilitate tumor cell metastasis by promoting cancer cell metabolic plasticity, and it enhances tumor cell resistance to chemotherapeutic agents [57].

Three general forms of autophagy are recognized: macroautophagy, chaperone-mediated autophagy (CMA), and microautophagy [58]. Macroautophagy, CMA, and microautophagy share some common upstream regulators, while they also exhibit unique regulatory mechanisms and downstream effectors that ensure their exquisite control [59]. This latter point is particularly important, since deregulated autophagy can cause increased cell stress, compromised fitness, and, in extreme situations, autophagic cell death [58]. Below, we discuss the basic genetic and biochemical mechanisms that underlie these three types of autophagy, and we outline their general regulation by upstream signaling pathways.

### 3.2. Overview of Macroautophagy

The genetic and biochemical control of macroautophagy (hereafter referred to as autophagy) has been characterized extensively using a variety of model organisms. These studies have revealed that autophagy is highly conserved across species [55], with many of the genes controlling each stage of the autophagy process having been originally identified in non-mammalian models such as budding yeast [60]. Autophagy can be broken down into four general steps that include (1) phagophore initiation; (2) phagophore expansion and closure to form the autophagosome; (3) autophagosome fusion with the lysosome; and (4) cargo degradation and nutrient efflux [58]. Each step is subject to specific regulatory control to prevent deregulated autophagy and the detrimental effects associated with extensive self-digestion, including autophagy-induced cell death. Autophagy initiation initially involves the activation of the ULK1 complex that consists of the ULK1 kinase and the ATG13, RB1CC1, and ATG101 subunits. ULK1 activation occurs in response to altered environmental signals, typically nutrient and/or energy depletion (discussed below). Activated ULK1 then signals to the phosphatidylinositol 3-kinase (PtdIns3K) complex that includes the PIK3C3/VPS34 kinase; the PIK3R4/VPS15 regulatory factor; and the additional BECN1/ATG6, NRBF2, and AMBRA1 subunits. This activated PtdIns3K kinase will phosphorylate lipid membrane phosphatidylinositols derived from multiple potential membrane sources, including from the ER, plasma membrane, ATG9-containing vesicles, and, potentially, endosomes [61]. The resulting phosphatidylinositol-3-phosphate (PtdIns3P) facilitates the formation of the double-membrane phagophore structure, and it functions as a docking site for the PtdIns3P binding proteins WIPI2 and ZFYVE1/DFCP1. The expansion of this initial phagophore structure then involves signaling through two ubiquitin-like regulatory pathways that have both shared and distinct components. The first pathway involves the conjugation of the ubiquitin-like ATG12 protein to ATG5. This conjugation is mediated by ATG7, which has similarity to E1 ubiquitin conjugating enzymes, and by ATG10 that resembles E2-like ubiquitin conjugating enzymes. Separately, the ATG8 family of proteins, including MAP1LC3 and the GABARAP family members, are proteolytically processed at their C-terminus by ATG4 to expose a C-terminal glycine residue. The processed LC3 is then conjugated to phosphatidylethanolomine (PE) in the phagophore membrane (referred to as lipidation) via the activity of ATG7, the E2-like enzyme ATG3, and the E3-like enzyme complex consisting of ATG12-ATG5-ATG16L1. The ATG12-ATG5-ATG16L1 complex is tethered to the growing phagophore membrane via interactions between ATG16L1 and WIPI2. The growing phagophore will eventually fuse to form a double-membrane autophagosome, and the ATG factors are removed. The resulting autophagosome either interacts with endosomes to form an amphisome that then fuses with the lysosome or it directly merges with the lysosome to form an autolysosome. Once deposited into the lysosome, the double-membrane structure and the enveloped cargo are broken down by a variety of resident degradative enzymes. The resulting amino acids, nucleotides, and other nutrients will then be effluxed from the lysosome into the cytoplasm to be recycled and used to support metabolism [61]. Autophagy can be non-specific, where random cytoplasmic constituents are degraded through the pathway, but it also can be selective, where only highly specific cargos are targeted for degradation. A wide variety of selective autophagy pathways have been documented, including mitophagy, lipophagy, proteaphagy, ribophagy, and many others [59,62]. Determining how these selective autophagic pathways are regulated, their impact on cell physiology, and their contribution to a variety of diseases including cancer, remains an active area of investigation [59]. For many of these pathways, ubiquitin-dependent signaling plays an integral role in marking specific cargos for degradation. In general, ubiquitinated cargo can be bound by proteins that contain ubiquitin-binding motifs coupled with LC3 interacting regions (LIRs). The bound cargo is then targeted into the growing phagophore through interactions with lipidated LC3. While ubiquitinated cargos can specify cargo targeting through the autophagy pathway, selective autophagy may also occur independently of a currently known role for ubiquitin signaling [63].

### 3.3. Chaperone-Mediated Autophagy (CMA)

The most distinctive difference between autophagy and either CMA or microautophagy (below) is that neither CMA nor microautophagy involves autophagosome formation and fusion with the lysosome for cargo degradation. Instead, CMA and microautophagy cargo capture occurs directly at the lysosomal surface through very distinctive mechanisms [64,65]. For CMA, individual proteins that carry exposed KFERQ or KFERQ-like sequences are recognized by the chaperone HSC70 and associated co-chaperones. Approximately 40% of the mammalian proteome has KFERQ-like sequences or the potential to generate KFERQ-like sequences through protein post-translational modification [64]. HSC70, bound to the KFERQ-containing cargo, then binds the lysosomal LAMP2A transmembrane oligomeric complex. To translocate the cargo through LAMP2A and into the lysosomal lumen, HSC70 and associated co-chaperones unfold the cargo and then thread it through LAMP2A. Cargo transit through LAMP2A also requires lysosomal lumenal HSC70, which is necessary for complete cargo translocation into the lumenal space [64]. Once the cargo reaches the lysosomal lumen, various proteases degrade it, and the released amino acids are then recycled back into the cytoplasm through the function of resident lysosomal nutrient efflux factors. Intriguingly, whereas both autophagy and microautophagy occur in most, if not all, eukaryotes, CMA is documented to occur only in mammalian cells and birds, suggesting this specific autophagy pathway is a late evolutionary invention [64,66]. 

### 3.4. Microautophagy

As mentioned above, microautophagy bypasses the requirement for autophagosome formation and fusion with the lysosome. Instead, cargos are captured directly by lysosomal or endosomal membranes via a mechanism involving either membrane protrusion or invagination [65]. This process can occur either in a non-selective or selective fashion. While some core autophagic factors may contribute to aspects of microautophagy, at least in some organisms such as yeast, microautophagy pathways may not depend on the ATG5 and ATG7 pathways [65,67,68]. Microautophagy can be subdivided into either fusion or fission microautophagy, which is regulated by distinct effectors. Fusion microautophagy uses some components of the core autophagy machinery and SNARE proteins [69]. This process is less well defined than fission microautophagy, which is independent of the core autophagy machinery but requires ESCRT factors [70]. Numerous specific microautophagy pathways exists, including micromitophagy, micronucleophagy, and microproteaphagy, among others [65]. 

## 4. Autophagy in GBM

### 4.1. The Role of Autophagy

The role of autophagy in cancer is somewhat controversial, as both pro-tumorigenic and anti-tumorigenic roles have both been reported for autophagy in the literature [71]. Nonetheless, autophagy emerges as a double-edged sword that plays a complex and context-dependent role in tumor development and cancer therapy [72]. In healthy cells, autophagy plays a tumor-suppressive role by maintaining normal homeostasis. In contrast, in cancer cells, autophagy functions in either a tumor-promoting or a suppressive role, which may reflect the tumor stage, the tumor microenvironment, and the heterogeneity of cancer stem cells in the tumor. For example, autophagy suppresses primary tumor growth, but it becomes necessary to support the elevated metabolic demand for tumor progression, and autophagy promotes multiple steps in the tumorigenic process [73]. The constitutive activation of the mTOR signaling pathway in GBM not only impairs basal autophagy but also enhances the proliferation and the stemness of GSCs [74]. In agreement with these findings, the pharmacological inhibition of mTOR induces autophagy and reduces the invasive potential of GSCs, suggesting that mTOR hyperactivation sustains GSC metabolism by means of suppressing autophagy [75]. In addition, increased autophagy has been associated with both tumor survival and chemoresistance in GBM [76], as shown in Figure 1.

Autophagy has been reported to be cytoprotective and to promote cancer progression by helping cancer cells to overcome chemotherapy-induced programmed cell death, as shown in Figure 2. For example, downregulation of autophagy effectors such as Atg7, Atg13, and ULK1 was associated with a significant reduction in glioma tumorigenesis [77]. Conversely, autophagy stimulation was found to reduce the migration and invasion of GBM cell lines [78]. High levels of the autophagy markers microtubule-associated protein 1A/1B-light chain 3 (LC3) and Beclin-1 were shown to be correlated with a better survival in GBM patients [79,80]. However, under the hypoxic conditions that are characteristic of the brain tumor microenvironment, the induction of autophagy upregulation was described both as a compensatory response protecting cancer cells from hypoxia [81] and as a potential death mechanism [82]. These findings have led to studies to determine whether modulating autophagy may sensitize brain tumor cells to chemotherapy or confer chemoresistance. For instance, the DNA alkylating agent TMZ, which is the front-line chemotherapy in GBM, was shown to induce a cytoprotective autophagy, which was suggested as a mechanism of therapeutic resistance [83]. Consistent with this approach, the inhibition of autophagy by treatment with the lysosomal inhibitor chloroquine enhanced the TMZ sensitivity of GBM cells [84]. Treatment of GBM with radiotherapy and TMZ induces autophagy that sustains tumor cell survival, and thus contributes to treatment resistance and recurrence [85]. Thus, it is not surprising that treatment with autophagy inhibitors chloroquine and hydroxychloroquine sensitized GBM cells to cytotoxic drugs [85,86]. Based on the mounting data, the therapeutic manipulation of autophagy, if finely tailored to brain tumors, remains a reasonable approach to explore (Figure 1). 

### 4.2. The Role of Autophagy in Cancer Stem Cells

In this section, we will focus our attention on the role of autophagy in GSCs, because cancer stem cells (CSCs) present a major challenge in the field due to their inherent chemoresistance and their ability to repopulate the tumor after surgery and treatment. The discrepancies in the literature regarding autophagy’s role in GBM are at least partly reflective of the contextual-dependent role autophagy has in maintaining the GSC stem-like state and metabolism, as well as its role in controlling GSC proliferation and differentiation. As GSCs have similar characteristics to CSCs isolated from other tumor types [87], studies examining the role autophagy has in these other CSCs may provide some insight into how autophagy functions in GSCs. Consistent with such studies, autophagy in both CSCs and GSCs is thought to contribute substantially to their inherent resistance to chemotherapy [88]. Autophagy is induced in response to both nutrient stress and other microenvironmental stressors including hypoxia. Solid tumors present a nutrient-poor, hypoxic environment [89], and this type of intratumoral niche can influence GSC maintenance and function [90]. Autophagy upregulation thus provides GSCs with a mechanism for adapting to this harsh tumor microenvironment. When these metabolic stress adaptations are combined with the intrinsically low proliferation rate of GSCs, these cells become inherently resistant to traditional chemotherapeutics, such as TMZ, relative to the other more differentiated and proliferative tumor cells [88]. Understanding how autophagy contributes to the biological adaptations of GSCs would have clinical benefit as it may be possible to inhibit the autophagy response in GSCs to limit its role in chemotherapy resistance, which would have potentially positive outcomes for patients.

### 4.3. Role of mTORC1-Dependent Autophagy Control in GSCs

mTORC1 signaling is essential for normal neural stem cell (NSC) maintenance, proliferation, and differentiation, while deregulating mTORC1 signaling disrupts NSC differentiation and maturation [91]. Not surprisingly, mutations that deregulate the PTEN-PI3K-Akt-mTORC1 signaling axis are common drivers in GBM [92,93]. This signaling deregulation is associated with increased tumor grade, maintaining GSC pluripotency, restricting GSC differentiation and proliferation, and mediating GSC radioresistance [92]. One significant consequence of PTEN-PI3K-Akt-mTORC1 deregulated signaling is autophagy suppression [61], and this altered GSC autophagy response is thought to contribute to GBM pathogenesis [92]. Intriguingly, endothelial secreted factors have been reported to sustain the GSC niche by activating GSC mTORC1 signaling, while inhibiting mTORC1 in GSCs prevents GSC expansion [94]. Previous studies also have indicated that mTORC1 inactivation, and the consequent upregulation of autophagy, restricts GSC stemness to facilitate GSC differentiation and inhibition of their tumorigenic properties [95]. While these mTor-specific effects are often conflated with autophagy induction, some of these mTORC1 dependent effects could be autophagy-independent and due to other aspects of mTORC1 signaling. For example, studies in normal NSCs have demonstrated that mTORC1 negatively regulates 4E-BP1 activity, which represses cap-dependent translation when upstream mTORC1 is inactivated, to control NSC maintenance and function. The inhibition of mTORC1, or 4E-BP1 constitutive activation, prevents NSC differentiation and reduces lineage expansion [96]. These results, at least in NSCs, indicate that mTORC1 inhibition has effects on the NSC niche that could be translatable to its role in GSCs and would be independent of autophagy. 

Related to the above issue, mTORC1 inhibitors are used frequently to induce autophagy in GSCs to study autophagy’s role in GSC biology. However, attributing any observed effect on GSCs solely to autophagy under these conditions should be met with some skepticism as most downstream mTORC1-regulated anabolic processes will also be impacted [97]. Even autophagy-inducing agents that block lysosomal-dependent protein degradation, such as chloroquine, have the potential to alter additional signaling pathways that impact cell metabolic homeostasis. This possibility is further supported by the increasing recognition that the lysosome functions as a metabolic signaling hub [98] and that key lysosomal regulators such as the V-ATPase H^+^-pump not only activate mTORC1 but also other core signaling pathways including PKA [99,100]. Distinguishing these additional effects that mTORC1 inhibition has on GSC biology from the contribution that autophagy itself makes will require more refined and selective genetic approaches. These approaches should uncouple mTORC1 inhibition and its repression of cell anabolic processes from its ability to induce autophagy. When applied to GSC studies, such approaches should allow the deconvolution of how mTORC1 inhibition and autophagy activation impact GSC activity compared to the role mTORC1 suppression only has on anabolic transcription, translation, and metabolism.

### 4.4. Recent Studies of Autophagy in GSC Regulation

Conflicting data do exist regarding autophagy’s role in GSCs. Recent work has provided evidence that autophagy inhibition, using shRNA-mediated silencing of Beclin1 or ATG5 expression, increases the expression of GSC stem cell markers while also promoting GSC proliferation and clonogenicity [101]. Although autophagy is thought to mediate resistance to chemotherapeutic agents like TMZ, in this study, Beclin1 inhibition failed to sensitize GSCs to TMZ-dependent apoptosis relative to control GSCs [101]. A mechanistic understanding of how autophagy inhibition in this experimental model enhances the GSC phenotype, or why it fails to sensitize GSCs to TMZ, remains unclear. Additionally, genetic analyses of GBM patient samples have found that a large percentage of GBM has deregulated signaling through the PTEN-PI3K-Akt-mTORC1 axis, which is often due to mutations in the PTEN tumor suppressor [102]. GSCs with PTEN-inactivating mutations increase mTORC1 signaling, which suppresses autophagy. However, this mTORC1 deregulation comes at a fitness cost that simultaneously creates a synthetic dependency on proteasome-mediated protein degradation to maintain proteostasis [103]. Proteasome inhibition in PTEN-null GSCs substantially decreases their viability, while simultaneous inhibition of both the proteasome and autophagy in PTEN-expressing GSCs causes cytotoxicity [103]. This work further underscores the importance for GSCs to maintain a basal level of proteostasis under genetic conditions that deregulate the PI3K-mTORC1 signaling pathway. However, this adaptive response makes these GSCs much more sensitive to combinatorial agents targeting proteostasis. This vulnerability could be a highly specific way to therapeutically target the GSC compartment in GBM while sparing bystander cytotoxicity. In line with this therapeutic approach, pharmacological targeting of the autophagy-activating ULK1 kinase in combination with tyrosine kinase inhibitors also exhibits an enhanced antileukemic effect in chronic myelogenous leukemia, which is a tumor propagated by the population of leukemic stem cells [103]. Therefore, the combinatorial inhibition of receptor tyrosine kinase pathways with autophagy inhibition may provide a common paradigm for therapeutic treatment for a subset of GBM and other tumor types. Consistent with this approach, the pharmacological inhibition of ULK1 suppresses STAT3-dependent autophagy and induces apoptosis in GBM cells [104]. Thus, combining STAT3 inhibition with mTOR inhibitors may be a novel approach to overcome chemoresistance and treat GBM by promoting autophagy.

Another recent study has identified an important role for mitophagy (autophagy-dependent degradation of mitochondria) as a tumor-suppressive mechanism in GBM. Deregulated PDGFR signaling is a frequent oncogenic driver in a subset of GBM, and one consequence of oncogenic PDGFR signaling is increased early growth response 1 (EGR1)-dependent transcription of the METTL3 RNA methyltransferase that methylates RNA to generate N^6^-methyladenosine (m^6^A) [105]. METTL3-dependent m^6^A mRNA modification can regulate gene expression post-transcriptionally, and in GSCs one key target of METTL3 is the mRNA that encodes optineurin (OPTN), which is a well-established activator of mitophagy [105]. PDGFR-mediated METTL3 upregulation decreases OPTN expression in patient-derived GSCs to maintain GSC function and GSC-dependent GBM tumorigenesis. Furthermore, METTL3 inhibition increases OPTN mRNA levels and enhances mitophagy to inhibit GSC-dependent GBM tumorigenesis, while OPTN overexpression also represses GBM tumorigenesis [105]. Consistent with OPTN having a tumor-suppressive role in GBM, analysis of tumor databases revealed that GBM patients with high OPTN expression have a significantly longer survival than patients with lower OPTN levels. Consistent with this pathway being a target for pharmacological intervention, the combined inhibition of METTL3 with PDGFR inhibitors resulted in enhanced GBM anticancer activity [105]. These preclinical studies indicate the possibility that this combinatorial approach may be clinically beneficial for GBM patients. However, clinical approaches that manipulate mitophagy should be viewed with some degree of caution, since additional work has indicated that increased mitophagy through different genetic means facilitates GBM and other tumor types, including non-small-cell lung cancer (NSCLC) [106,107]. Specifically in the case of NSCLC, CSCs are maintained by signaling through a mitophagy-dependent activation of the Toll-like receptor 9 (TLR9). TLR9 then stimulates Notch1 and AMPK kinase signaling to enhance mitochondrial metabolism and promote CSC expansion and tumorigenesis [107]. 

While these studies examined general autophagy in GSC biology, a recent study has focused on the contributory role CMA has in promoting GSC-dependent GBM tumorigenesis [108]. Specifically, LAMP2A is overexpressed in patient-derived GSCs and in GBM patient samples. Downregulating LAMP2A expression by shRNA-mediated knockdown reduces GSC proliferation and tumorigenicity. Increasing LAMP2A levels also enhances the expression of stem cell markers in the GSC phenotype, thus further supporting a role for enhanced CMA in maintaining the GSC phenotype and GBM tumorigenesis. Transcriptome and proteome analysis of GSCs with reduced LAMP2A expression found a reduction in factors mediating extracellular matrix interactions, and changes to pathways involved in mitochondrial function and immune-related pathways, including interferon signaling [108]. These data provide candidate downstream cellular pathways affected by CMA activity that may play a role in GSC maintenance and tumorigenesis. 

## 5. Crosstalk between STAT3 and Autophagy in GBM

Although these studies underscore the relevance of autophagy in GBM, comparatively little is known about the function of deregulated STAT3 signaling in GBM and its impact on autophagy. The pharmacologic inhibition of either JAK2 (using SAR317461) [109] or STAT3 (using AG490) [110] stimulates autophagy in GBM cells. STAT3 inhibits autophagy not only by upregulating anti-autophagy genes but also by downregulating pro-autophagy genes [111]. An inverse correlation between phosphorylated STAT3 and the cellular levels of a stimulator of autophagy, Beclin1, has also been observed in GBM [111]. A recent study using GBM cells that had STAT3 knocked out by CRISPR/Cas9 gene editing examined the specific STAT3-dependent signaling mechanisms that modulate autophagy [104]. Utilizing STAT3 knockout (STAT3-KO) GBM cells and STAT3-KO cells restored with wild-type STAT3 or mutants deficient in Y705ph or S727ph showed that deregulated STAT3 activation in GBM cells suppressed autophagy as determined by the phosphorylation of AMPKα and Unc-51-like kinase 1 (ULK-1). While KO of STAT3 increased AMPKα phosphorylation, the restoration of STAT3 expression with wild-type STAT3 reduced AMPKα phosphorylation. In contrast, restoration with either phosphorylation-deficient mutant resulted in high levels of AMPKα phosphorylation. In line with the findings for AMPKα, STAT3-KO in GBM resulted in high ULK-1 phosphorylation, while restoration with wild-type STAT3 reduced ULK-1 phosphorylation. Restoration with either phosphorylation-defective mutant of STAT3 did not result in increased ULK-1 phosphorylation (Figure 2). In addition, the treatment of GBM cells with bafilomycin, an inhibitor of the vacuolar ATPase that prevents the fusion of lysosomes and autophagosomes, provided further evidence that STAT3 suppressed autophagy. While treatment with bafilomycin significantly increased LC3-II levels in STAT3-KO GBM cells, LC3-II levels were markedly reduced in bafilomycin-treated GBM cells restored with wild-type STAT3. Most importantly, GBM cells restored with either phosphodeficient STAT3 mutant showed both high basal and bafilomycin-induced LC3-II levels. These results were further confirmed by immunolocalization studies to detect LC3 and p62 puncta formation, which is the classical measurement of autophagy flux. These results suggest a model in which both serine and tyrosine phosphorylation of STAT3 are responsible for the inhibition of autophagy in GBM cells through the inhibition of an AMPKα/ULK1 signaling pathway. The pharmacologic inhibition of mTORC1 with everolimus stimulates autophagy, while inhibiting ULK1 with MTY68921 or siRNA knockdown of ULK1 inhibits autophagy and induces apoptosis in STAT3-KO cells. Together, these studies demonstrate that a STAT3-dependent pathway suppresses autophagy in GBM cells. Furthermore, using a combination of STAT3 and mTOR inhibitors to promote autophagy may be a novel approach to overcome chemoresistance and treat GBM.

## 6. Conclusions and Future Perspectives

The role that autophagy plays in GBM pathogenesis is complex and at times controversial. For example, autophagy is activated in glioma cells and promotes apoptosis in response to various cellular stressors including treatment with chemotherapeutic drugs and hypoxia. Based on such findings, inhibitors of lysosomal proteolysis that block autophagy have shown efficacy in some preclinical and clinal studies. During the advanced stage of the disease, the induction of autophagy plays an important role in the survival of GBM cells as it provides metabolic support and prevents cellular senescence and promotes chemoresistance. Similarly, to the complex role that autophagy plays in GBM in general, the role that autophagy plays in GSC is equally complex as it promotes the stem-like properties of GSCs and GSC invasiveness. Future studies are needed to unravel the potential role that the modulation of autophagy in GBM can play in inhibiting GBM tumorigenesis and in overcoming the therapeutic resistance of GBM to chemotherapy and radiotherapy.

## Figures and Tables

**Figure 1 pharmaceuticals-16-00671-f001:**
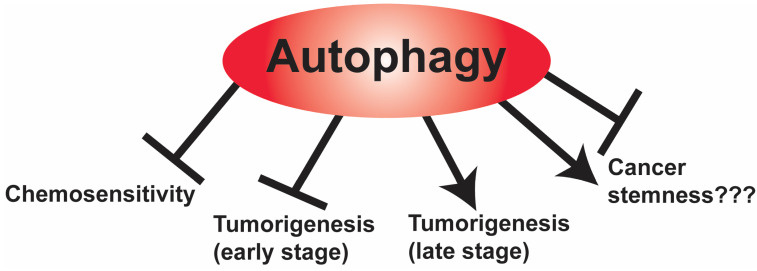
The complex roles of autophagy in GBM. In GBM cells, autophagy promotes GBM resistance to chemotherapy. Autophagy inhibits the early stages of GBM tumorigenesis but promotes the later stages of tumorigenesis. The role of autophagy in GBM cancer stemness is also complex, with reports suggesting that it promotes cancer stemness, while other studies suggest that it promotes differentiation of GBM cancer stem cells.

**Figure 2 pharmaceuticals-16-00671-f002:**
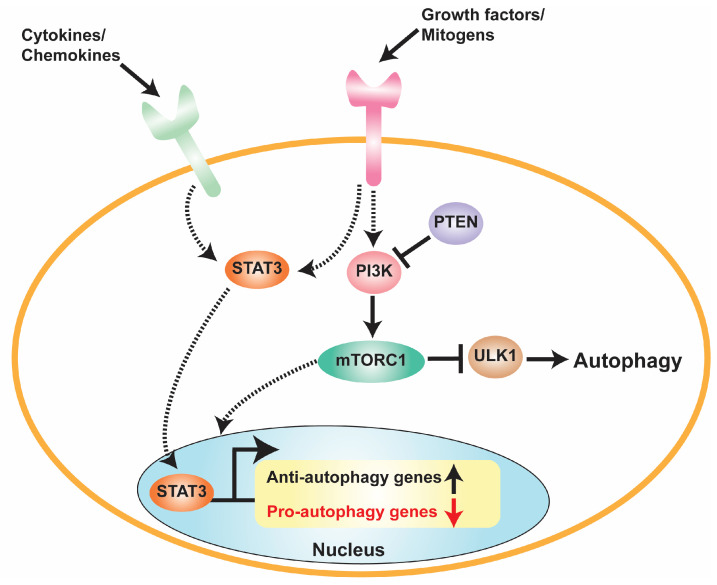
Schematic showing the molecular mechanisms affecting autophagy in GBM. In GBM cells, chemokines, cytokines, growth factors, and mitogens induce STAT3 activation through STAT3 Y705 tyrosine phosphorylation. Activated STAT3 then translocates into the nucleus to induce the expression of anti-autophagy genes and suppress the expression of pro-autophagy genes. Growth factors and mitogens also activate the PI3K pathway, which subsequently leads to mTORC1 activation and the inhibition of ULK1 signaling to inhibit autophagy. mTORC1 also regulates the STAT3 activation and the expression of pro- and anti-autophagy genes.

## Data Availability

Data sharing not applicable.

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
