# Peer review of "The STAT3-Regulated Autophagy Pathway in Glioblastoma"

_pharmaceuticals, 2023, doi:10.3390/ph16050671_

Round 1

Reviewer 1 Report

Thanks a lot for your invitation to review this manuscript "The STAT3 regulated autophagy pathway in glioblastoma" (pharmaceuticals-2329626). I think that this work could be accepted for publication after major revision. The detailed comments were shown as follows:

1. Introduction should be summarized. 

2. for the readers better understand, it will be better to add some graphical schema such as pathways maps to each section of the manuscript.

3. The current treatment methods for glioblastoma should be discussed and compared.

4. the STAT pathway activator main drugs should be discussed.

Author Response

Reviewer #1

  1. Introduction should be summarized. We have reorganized and revised the introduction to improve clarity.
  2. For the readers better understand, it will be better to add some graphical schema such as pathways maps to each section of the manuscript. We have added two Figures that show various pathways involved in autophagy in GBM.
  3. The current treatment methods for glioblastoma should be discussed and compared. In the revised manuscript, section 1.2 described the present clinical treatments for GBM.
  4. the STAT pathway activator main drugs should be discussed. The major STAT3 inhibitor for GBM are descried in section 2 (lines 160-167).

Reviewer 2 Report

As well-known that autophagy is a double-edged sword and may have anti-tumor properties under certain conditions. The present review try to indicate that the role of the STAT3 transcription factor in autophagy with GBM as well. The highlights appear reasonable; however, there exists a potential big problem with lacking clear figures and crosstalk between STAT3 and autophagy signaling pathways in GBM:

1.      In section 2. The STAT3 signaling pathway in GBM, better to draw Figure 1 to make more clear.

2.      Same as above in section 3. Autophagy needs Figure 2a and section 4. Autophagy in GBM (Figure 2b), what are their features and unique for GBM with autophagy

3.      Better add a new section 5 of crosstalk between autophagy and STAT3 signaling pathway in GBM, and also draw Figure 3 as indicated.

4.      Move section 5. Conclusions and future perspectives to section 6

Author Response

Reviewer #2

  1. In section 2. The STAT3 signaling pathway in GBM, better to draw Figure 1 to make more clear. Figure 2 in the revised manuscript is a schematic showing the role of STAT3 in autophagy.
  2. Same as above in section 3. Autophagy needs Figure 2a and section 4. Autophagy in GBM (Figure 2b), what are their features and unique for GBM with autophagy. We have added two additional Figures that show various pathways involved in autophagy in GBM.
  3. Better add a new section 5 of crosstalk between autophagy and STAT3 signaling pathway in GBM, and also draw Figure 3 as indicated. We have added a new section 5 as requested entitled Crosstalk between STAT3 and autophagy in GBM. Figure 2 in the revised manuscript is a schematic showing the role of STAT3 in autophagy.
  4. Move section 5. Conclusions and future perspectives to section 6. As requested, we have moved Conclusions and future perspectives to section 6.

Reviewer 3 Report

n the present review titled ‟The STAT3 regulated autophagy pathway in glioblastoma”, authors meticulously discussed STAT3 signaling in glioblastoma (GBM), glioblastoma stem cells (GSC) and its role in  macroautophagy. This review speculates possible therapeutic target of stat3 mediated autophagy for GBM and GSC. 

Minor Comment: 

The present manuscript described in detail various autophagy processes, stat3 signaling mechanisms. The manuscript would be more reader friendly with graphical summarization of all these mechanisms and possible approaches to target GBM and GSC.    

Author Response

Reviewer #3

  1. The manuscript would be more reader friendly with graphical summarization of all these mechanisms and possible approaches to target GBM and GSC.   As requested, we added two Figures that show various pathways involved in autophagy in GBM and GSCs.

Round 2

Reviewer 1 Report

Accepteble in the current version.

Reviewer 2 Report

No further comment. The present version has been sufficiently improved to warrant publication in Pharmaceuticals